# Cellular Therapies in Chronic Lymphocytic Leukemia and Richter’s Transformation: Recent Developments in Chimeric Antigen Receptor T-Cells, Natural Killer Cells, and Allogeneic Stem Cell Transplant

**DOI:** 10.3390/cancers15061838

**Published:** 2023-03-18

**Authors:** Catherine C. Coombs, Saumya Easaw, Natalie S. Grover, Susan M. O’Brien

**Affiliations:** 1Division of Hematology and Oncology, Department of Medicine, University of California Irvine, Orange, CA 92868, USA; 2Atrium Health, Carolinas Hospitalist Group, Charlotte, NC 28203, USA; 3Division of Hematology, Department of Medicine, University of North Carolina at Chapel Hill, Chapel Hill, NC 27599, USA

**Keywords:** chronic lymphocytic leukemia, Richter’s transformation, Richter’s syndrome, allogeneic stem cell transplant, chimeric antigen receptor T-cells, CAR-T

## Abstract

**Simple Summary:**

Here, we present a review of cellular therapies that have been used in both chronic lymphocytic leukemia and Richter’s transformation. We will review the data behind allogeneic hematopoietic stem cell transplantation in the modern era. Subsequently, we will discuss existing and emerging strategies for chimeric antigen receptor-based approaches.

**Abstract:**

Cellular therapies can be viewed as both the newest and oldest techniques for treating chronic lymphocytic leukemia (CLL) and Richter’s transformation (RT). On one hand, allogeneic hematopoietic stem cell transplantation (alloHSCT) has been available for decades, though its use is diminishing with the increasing availability of effective novel targeted agents, especially in CLL. Among newer techniques, chimeric antigen receptor T-cells (CAR-T) have demonstrated astounding efficacy in several hematologic malignancies, leading to FDA approval and use in clinical practice. However, though CLL is the earliest disease type for which CAR-T were studied, development has been slower and has yet to lead to regulatory approval. Owing partially to its rarity but also due to the aggressive behavior of RT, CAR-T in RT have only been minimally explored. Here, we will focus on the applications of cellular therapies in CLL and RT, specifically reviewing more recent data related to alloHSCT in the novel-agent era and CAR-T cell development in CLL/RT, focusing on safety and efficacy successes and limitations. We will review strategies to improve upon CAR-T efficacy and discuss ongoing trials utilizing CAR-T in CLL/RT, as well as emerging technologies, such as allogeneic CAR-T and natural killer CAR (CAR NK) cells.

## 1. Introduction

Cellular therapies have been in use for decades for treating hematologic malignancies, including chronic lymphocytic leukemia (CLL) and Richter’s transformation (RT). The earliest cellular therapy was allogeneic hematopoietic stem cell transplantation (alloHSCT), though the use of this technique has been declining with the broader availability of novel targeted therapies, including covalent BTK inhibitors (BTKi), venetoclax, and to a lesser extent, phosphoinositide 3-kinase inhibitors (PI3Ki), which will collectively be referred to as novel agents in this review. Chimeric antigen receptor T-cells (CAR-T) are T-cells that have been genetically modified to express a receptor that recognize a pre-specified target. Among hematologic malignancies, the most common target has been CD19 for which CAR-T cells have been FDA approved for use in several lymphoid malignancies, including B-cell acute lymphoblastic leukemia (ALL) and B-cell non-Hodgkin’s lymphomas (NHL) including diffuse large B-cell lymphoma (DLBCL), mantle cell lymphoma (MCL), and follicular lymphoma (FL) [1,2]. There have also been two FDA-approved products for use in multiple myeloma (MM) that target B-cell maturation antigen (BCMA) (idecabtagene vicleucel and ciltacabtagene autoleucel) [3,4]. CAR-T remain investigational for many other malignancies [5,6].

CLL was the first malignancy for which CAR-T were administered therapeutically, but subsequent development has been slower, primarily due to challenges related to the underlying immune milieu that occurs with this disease [7,8]. Ongoing strategies that focus on improving the efficacy of cellular therapies including CAR-T are of paramount importance, given the need for effective therapies following failure of novel agents, including covalent BTK inhibitors and venetoclax; this group represents a significant unmet need in CLL [9]. This group has been termed “double exposed” or “double refractory” based upon whether patients simply had prior exposure to covalent BTKi and venetoclax (with a portion discontinuing due to intolerance) versus whether they were refractory to both agents. Retrospective series have shown a dismal median overall survival (OS) of 3.6 months for patients who are “double refractory” [10] or 5.6 months for those “double exposed” [11]. Another study emphasized short progression-free survival (PFS) for available subsequent therapies, including chemoimmunotherapy and PI3Ki (three and five months, respectively) for patients who were “double exposed” [12]. RT also remains a major challenge therapeutically, with extremely poor outcomes even in the novel-agent era, with median OS of less than one year [13,14].

## 2. What’s “New” in AlloHSCT for CLL and RT: Focus on Patients with Prior Novel-Agent Exposure

### 2.1. AlloHSCT in CLL

Transplantation in CLL has been on the decline in recent years, largely due to the availability of highly effective targeted agents such as covalent BTKi and venetoclax [15,16]. Nonetheless, alloHSCT can provide long-term disease-free survival and potentially a cure, demonstrated by both registry data and retrospective analyses (Figure 1) [15,17,18]. Improved outcomes have been noted in patients undergoing reduced-intensity conditioning (RIC) prior to alloHSCT [18,19,20]. Further, donor availability has been broadened through the use of haploidentical donors and post-transplant cyclophosphamide [21].

Data for the use of alloHSCT in the novel-agent area are limited. One study examined the feasibility of the revised European Research Initiative on CLL(ERIC)/European Society for Blood and Marrow Transplantation (EBMT) recommendations, which propose alloHSCT for patients (level 1) with a low-transplant risk but high disease risk (*TP53* abnormalities and/or del(11q), though del(11q) was removed on revisions) who are relapsed/refractory to chemoimmunotherapy but are responding to first novel agent, or patients (level 2) with disease that is relapsed/refractory to both chemoimmunotherapy and first novel agent and may be higher risk with respect to alloHSCT (i.e., mismatched donor, comorbidities) [22]. Of 21 patients referred to their institution who met eligibility to trigger a donor search, 19 met level 1 criteria, of whom 11 underwent alloHSCT; nine of these patients are in ongoing remission at time of publication. The level 2 patients included a mix of nine patients who had either progressed from level 1 or de novo; seven underwent alloHSCT and four were in ongoing remission [22]. Overall, these data suggest the feasibility of their algorithm but whether it should be broadly applied in practice is a separate consideration, especially given the availability of multiple novel agents where one can work following progression on the other [23,24,25]. Another limitation to this algorithm is the requirement for progression on chemoimmunotherapy which is being used less frequently in modern practice due to the OS advantage of novel agent approaches [26].

A multicenter study of 65 patients with CLL who underwent alloHSCT following exposure to at least one novel agent demonstrated that outcomes were reasonably favorable with two-year OS and PFS of 81% and 63%, respectively (CLL diagnosed at median age of 50 with median age at time of alloHSCT of 60). When assessing for independent predictors of outcome, PFS did not differ in patients with poor-risk disease characteristics, prior novel agent exposure, or presence of complete remission (CR) vs. partial remission (PR); higher hematopoietic cell transplantation-specific comorbidity index led to shorter PFS [27]. Similarly, Mukherjee et al. analyzed outcomes for 49 patients transplanted at MD Anderson following novel-agent exposure, including 17 CLL patients (in addition to 14 patients with RT and 18 with other lymphomas) who achieved three-year OS and PFS of 68% and 59%, respectively [28]. At the Dana Farber Cancer Institute (DFCI), 30 patients (median age = 60) underwent alloHSCT following novel agent use, and three-year OS and PFS were 87% and 69%, respectively, which compared favorably to the pre-novel agent comparator group with three-year OS and PFS of 69% and 58%, respectively [29]. Lahoud et al. reviewed outcomes of 58 CLL patients, including 23 RT patients, who underwent RIC alloHSCT at Memorial Sloan Kettering Cancer Center, noting five-year PFS and OS of 40% and 58%, respectively, with comparable outcomes for patients with and without RT. Use of novel agents, received by 19% of patients in this study, also did not impact alloHSCT outcomes [30]. A subset of patients (*n* = 19) in another study received both alloHSCT and a novel agent (either before and/or after alloHSCT) and experienced favorable survival outcomes (three-year OS 83%) [31].

The majority of novel-agent exposed patients in these series had only a single novel agent prior to alloHSCT; the data for “double exposed” CLL patients are sparse. However, in a series of 125 “double exposed” CLL patients, alloHSCT appears to be an effective approach with an overall response rate (ORR) of 76.5% and median PFS of 11 months [12].

### 2.2. AlloHSCT for RT

Kim et al. focused on alloHSCT outcomes for RT patients in a series of 28 patients from DFCI [32]. Median number of therapies for CLL and RT prior to alloHSCT was three: one for CLL and two for RT; where nine patients had received novel agents in addition to chemoimmunotherapy prior to alloHSCT (four for CLL and five for RT). Four-year OS and PFS were 53% and 39%, respectively [32], similar to a recently published meta-analysis of alloHSCT in RT [33]. A large analysis using the Center for International Blood and Transplant Research (CIBMTR) registry examined outcomes for both alloHSCT and autologous transplantation (autoHSCT) for patients with RT in the novel-agent era (DLBCL-type only, which accounts for approximately 90% of RT events). The autoHSCT cohort (*n* = 53) included more patients in CR (66% vs. 34%), whereas the alloHSCT cohort (*n* = 118) included more patients with 17p deletion (33% vs. 7%) and receipt of prior novel agents (39% vs. 10%). The three-year PFS and OS were 48% and 57% in the autoHSCT group, and 43% and 52% in the alloHSCT group, respectively [34]. Notably, these series did not distinguish between clonally-related and clonally-unrelated RT which significantly impact outcomes, though typically the majority are clonally-related events [35].

### 2.3. AlloHSCT Limitations

In addition to relapse rates noted above, toxicity remains a major barrier to more widespread use of alloHSCT in CLL and RT given the older age for the typical affected patient. Rates of non-relapse mortality (NRM) range from 7% (three-year), 10% (one-year), to 13% (two-year), though these are from heavily selected groups, who are typically significantly younger than the average CLL patient [27,28,29]. NRM may be a bit higher in RT patients undergoing alloHSCT, estimated between 10% (one-year though up to 25% for three-year in patients achieving CR or PR) and 29% (four-year), similarly from heavily selected series [32,34]. Consequently, the role for alloHSCT remains limited though may be considered in CLL patients with few or no standard of care or investigational options, and it likely remains the only curative approach for clonally related RT.

## 3. CD19 CAR-T Therapies Approved in Other Indications

There are four FDA-approved CD19 targeted CAR-T cells including lisocabtagene maraleucel (liso-cel), tisagenlecleucel (tisa-cel), brexucabtagene autoleucel (brexu-cel), and axicabtagene ciloleucel (axi-cel), with characteristics and approved indications depicted in Table 1. Among these, liso-cel has been studied the most widely in CLL. Turtle et al. reported on 24 patients with CLL, of whom 19 had progressed on ibrutinib and 6 had progressed on venetoclax. The ORR was 71% (17 of 24); 81% of patients with marrow involvement at baseline (17 of 21) having no detectable CLL by high-resolution flow on repeat marrow (seven of 12 tested cleared malignant IGH sequences) [36]. Cytokine release syndrome (CRS) was seen in 20 patients (83%) (18 with grade ½, one with grade 4 and one with grade 5) and eight patients (33%) developed neurotoxicity (five with grade 3, and one with grade 5), which appeared more common in patients with a higher percentage involvement of CLL in the baseline marrow [36].

The TRANSCEND CLL 004 trial enrolled patients with prior BTKi and ≥2 prior lines of therapy and included an arm with liso-cel monotherapy and an arm of liso-cel plus ibrutinib. Among enrolled patients on the liso-cel monotherapy arm, the product was successfully manufactured in 23 of 24 patients. One had RT following leukapheresis and was excluded from efficacy analysis. After a median follow up of 24 months, the ORR was 82% with 45% CR/CR with incomplete count recovery (Cri). Twenty patients were evaluable for MRD assessment, of whom 75% attained undetectable MRD (uMRD) in the blood and 65% in the marrow at a level of <10^−4^. CRS was seen in 74% of patients (grade ¾ in 9%), and 39% of patients had neurological events (grade ¾ in 22%). The median PFS was 18 months for all comers, though among the 11 double-exposed patients, the median PFS was 13 months [37].

**Table 1 cancers-15-01838-t001:** FDA-approved CD19 targeted CAR-T products, approved indications, characteristics, and whether products have been studied in CLL and/or RT.

CAR-T Product	Approved Indication(s)	Construct	Studied in CLL and/or RT
Lisocabtagene maraleucel (Breyanzi)	Adult patients with relapsed(rel)/refractory(ref) large B-cell lymphoma including diffuse large B-cell lymphoma (DLBCL) NOS, including DLBCL arising from indolent lymphoma, high-grade B-cell lymphoma, primary mediastinal lymphoma, and follicular lymphoma (FL) grade 3B who are refractory to firstline chemoimmunotherapy or relapsed within 12 months, who are rel/ref after firstline and are ineligible for HSCT due to comorbidities or age, or rel/ref after two or more lines of systemic therapy [38,39]	Anti-CD19 4-1-BB CD3z CD4:CD8 1:1	CLL [36,37,40] and RT [36,41]
Tisagenlecleucel (Kymriah)	Adult patients with rel/ref DLBCL after two or more lines of systemic therapy, including DLBCL NOS, high grade B-cell lymphoma, and DLBCL arising from FL, adult patients with rel/ref FL after two or more lines of systemic therapy, and young patients (up to age 25) with refractory or in second or later relapse B-cell precursor acute lymphoblastic leukemia (ALL) [42,43,44]	Anti-CD19 4-1BB CD3z	CLL [7,45,46,47]
Brexucabtagene autoleucel (Tecartus)	Patients with rel/ref mantle cell lymphoma and adult patients with relapsed or refractory B-cell precursor ALL [48,49]	Anti-CD19 CD28 CD3z Manufacturing step added to remove CD19 tumor cells	CLL [50]
Axicabtagene ciloleucel (Yescarta)	Adult patients with large B-cell lymphoma that is refractory to firstline chemotherapy or that relapses within 12 months of firstline chemotherapy, adult patients with rel/ref large B-cell lymphoma after 2 or more lines of systemic therapy, including DLBCL NOS, primary media-stinal large B-cell lymphoma, and DLBCL arising from FL, and FL after 2 or more lines of systemic therapy [51,52,53]	Anti-CD19 CD28 CD3z	CLL [54] and RT [55]

Tisa-cel (CTL019) led to a 57% ORR in a heavily pretreated CLL population (median of five prior therapies), with four CR and four PR; 23 patients were enrolled though only 14 were infused (of the nine who were not infused, three died from disease complications, four withdrew consent, and two were screen failures). Nine patients developed CRS (two with grade 3 and four with grade 4), and CRS was associated with clinical response [45]. Impressively, two patients treated with tisa-cel who attained CR were reported to have sustained remissions a decade later, with ongoing detectable CD4+ CAR T cells [56]. Subsequently, a randomized dose optimization study of tisa-cel was conducted; more responses were seen at the higher dose (5 × 10^8^) which was selected for further study [47]. Of 32 evaluable patients, the CR and ORR were 28% and 44%, respectively. With a median of 31.5 months of follow up, the median PFS was 40.2 months in patients who attained CR compared to one month to those who did not [46].

Long term follow-up of the study using FMC63-28Z (axi-cel uses the same CAR) included seven CLL patients (median of four prior therapies). CR and ORR were 63% and 88%, respectively; of the five CR, three were ongoing at extended follow up with a median duration of ongoing responses of 82 months. Median event-free survival (EFS) was 40.5 months [57]. Brexu-cel (KTE-X19) is being tested in ZUMA-8, a multicenter phase ½ trial in relapsed/refractory CLL [50]. A unique aspect of brexu-cel production is that malignant cells are removed during the manufacturing process, though it is unclear whether that step will lead to improved outcomes. ZUMA-8 included 15 CLL patients in four cohorts at varying doses, in patients with low tumor burden and in patients with ibrutinib as their last line of therapy (given up to 30 h prior to leukapheresis). At median follow up of 30.3 months, toxicity signals were consistent with prior brexu-cel studies, and objective responses were seen in seven of 15 patients, including two CR (seen in the low tumor burden group) [58].

## 4. Emerging Strategies to Improve CAR-T Efficacy

### 4.1. Ibrutinib and Other Combinations

Strategies to improve upon CAR-T efficacy have been explored, with the best studied being the addition of ibrutinib to CAR-T. Ibrutinib is an attractive partner given its favorable effects on T-cell function and number, improvement in proliferation of CAR-T cells, and attenuation of CRS [59,60,61]. TRANSCEND CLL 004 included a liso-cel plus ibrutinib arm, which has been presented in abstract form [40]. Nineteen patients with a median of four prior therapies received liso-cel with concurrent ibrutinib. Patients either needed to have prior progression on ibrutinib, presence of a BTK or PLCγ2 mutation, or high-risk features with more than six months of ibrutinib (with less than CR). Seventeen patients (89%) achieved undetectable MRD in the blood (15% or 79% in the marrow) by next-generation sequencing (≤10^−4^). The safety profile was manageable with 74% of patients having CRS though only one had grade 3 CRS, and six had neurological events (three were grade ≥ 3).

Gill et al. reported on a phase 2 trial examining the addition of autologous anti-CD19 humanized binding domain T-cells (huCART-19 or CTL119) to ibrutinib in patients not in CR despite six months of ibrutinib. Twenty patients were enrolled with 19 infused (one was excluded due to development of RT and lung adenocarcinoma). Six patients were on frontline ibrutinib at the time of enrollment. No bridging therapy was used. The three-month CR rate was 44% though 72% of patients had undetectable MRD (<10^−5^) at 12 months. Eighteen of the 19 infused patients developed CRS (grade 1/2 in 15 patients) and five developed neurotoxicity [62].

An additional study examined a different CD19-targeting CAR-T (JCAR014) in combination with ibrutinib. Nineteen patients were included with a median of five prior therapies. For those who received frontline ibrutinib, they must have progressed or not responded to it. ORR was 83% with 61% of patients achieving undetectable MRD in the marrow. CRS occurred in 74% of patients. However, CRS severity was lower with an accompanying lower concentration of CRS-associated cytokines when compared to patients receiving CAR-T without ibrutinib on the same clinical trial [63]. The combination was feasible in most, though ibrutinib dose reduction or discontinuation was needed in six patients, and one patient died from probable ibrutinib-associated cardiac arrhythmia during CRS [63].

T-cell exhaustion is well described in CLL and can manifest via increased expression of CD244, CD160, and PD1 [64,65]. Pembrolizumab, an anti-PD1 checkpoint inhibitor, appears safe though with modest efficacy in the post-CAR-T setting (25% ORR) in a study in 12 patients with large B-cell lymphoma (LBCL). However, in the patients who did respond, an expansion of the existing CAR-T product was observed [66]. Consequently, an attractive partner to CAR-T is a checkpoint inhibitor, which is being studied in combination with tisa-cel (with pembrolizumab, phase 1b PORTIA trial), axi-cel (with atezolizumab, ZUMA-6 trial) and liso-cel (with durvalumab, PLATFORM trial), both in the setting of relapsed/refractory aggressive B-cell NHL, with high response rates on preliminary reports [67,68,69]. Given the impact of T-cell exhaustion on CAR-T efficacy in CLL, combinations with PD-1 or PD-L1 inhibitors may be an attractive partner for future studies.

### 4.2. Novel CAR-T Targets and Constructs

Although CD19 has been a promising target in CLL, novel targets are still needed. Some patients develop resistance to CD19 CAR-T cells from antigen loss [36,70]. There have been several clinical trials evaluating CD20-directed CAR-T cells, given the universal expression and successful targeting of CD20 in mature B cell malignancies, including CLL. The preliminary results of an ongoing phase 1/2 clinical trial (NCT03277729) of a third generation CD20 CAR-T cells (MB-106) with both a 4-1BB and CD28 costimulatory domain in patients with relapsed refractory B-cell NHL and CLL have recently been reported [71]. This trial included patients who had previously received CD19 CAR-T cells. Sixteen patients had been treated including one patient with CLL. The overall response rate was 94% with 15 of 16 treated patients responding and 10 of 16 patients achieving a CR (62%), including the patient with CLL who had a negative PET and undetectable MRD in the peripheral blood and bone marrow by flow at day 28 post treatment. Seven patients (44%) had CRS and 1 patient (6.2%) had ICANS. There was no grade 3 or higher CRS or ICANS. These preliminary results are encouraging as we await further follow up of long-term responses.

There are also several other targets that have had promising preclinical results. CAR-T cells targeting B-cell activating factor receptor (BAFF-R), which is a B-cell specific marker, showed activity against human lymphoma and leukemia lines in vitro and in mouse models, and were active against CD19 negative targets, supporting further investigation of this target in clinical trials, including for relapsed disease post CD19 CAR-T cells [72,73]. CD37 also is highly expressed in B cell lymphomas, including CLL, and anti-CD37 CAR-T showed activity in mouse models, including against CD19 negative targets. There is currently a clinical trial of CD37 targeted CAR-T cells in hematologic malignancies including CLL (NCT04136275).

Another method of overcoming CD19 antigen escape is via CAR-T cells targeting more than one antigen. The most widely studied bispecific CARs have been CAR-T cells targeting CD19 and CD22, which is particularly promising in ALL [74,75]. However, in CLL, CD20 is a more relevant target. In a phase 1 trial, patients with relapsed/refractory B-cell NHL and CLL were treated with tandem bispecific CAR-T cells targeting CD20 and CD19 (LV20.19) [76]. Twenty-two patients were treated with an overall response rate of 82% and a CR rate of 64%. The ORR for the 12 patients treated at the highest dose level was 100%. The median duration of response for patients in CR was not reached at a median follow up of 10.1 months. One patient (5%) had grade 3–4 CRS and three patients (14%) had grade 3–4 neurotoxicity. Three patients with CLL were treated with two patients achieving CR and one patient achieving PR. Of note, all patients who relapsed did not lose CD19 antigen at time of relapse. Another study of tandem CAR-Ts targeting CD19 and CD20 in B-cell NHL and CLL had similar results, with an overall response rate of 79% and a CR rate of 71% among 28 patients treated [77]. The 12-month PFS was 64%. Further, 50% of patients had CRS with 14% grade 3, and no patients had grade 3 or higher ICANS. One CLL patient was treated on this trial and achieved a CR to therapy. These trials suggest promise in investigating dual targeting CAR-T cells.

Another challenge with CD19 CAR-T cells is the on target off tumor toxicity associated with depletion of normal B cells, leading to B cell aplasia and hypogammaglobulinemia. Therefore, there has been interest in evaluating targets with more limited expression on normal B cells to limit immunosuppression of treatment. Monoclonal B cells in a mature B-cell malignancy express immunoglobulins with either κ or λ light chains. Therefore, CAR-T cells targeting the κ light chain would recognize malignant κ-expressing B-cells but spare normal B cells that express the λ light chain. A phase 1 clinical trial treated patients with relapsed/refractory κ-expressing NHL, CLL, and multiple myeloma with CAR-T cells targeting the κ-light chain [78]. Nine patients with relapsed NHL or CLL were treated with responses seen in NHL patients, including two achieving a CR and one patient achieving a PR. Two patients with CLL were treated on the trial, resulting in one patient with no response and one patient stable disease. There were no CAR-T related toxicities. The majority of patients on this trial did not experience lymphodepletion, which may be a factor affecting the relatively low efficacy. There is currently a phase 1 clinical trial of CAR-T cells targeting the κ-light chain in patients with relapsed/refractory κ-expressing B-NHL and CLL (NCT04223765). In addition, preclinical studies have demonstrated antitumor activity of CAR-T cells targeting the λ-light chain while sparing κ-expressing normal B cells [79]. If found to be effective, this approach can preserve some normal B cell function in patients with mature B-cell malignancies treated with CAR-T cells.

Another target that has been investigated in CLL is receptor tyrosine kinase-like orphan receptor 1 (ROR1), which is also highly expressed in CLL but not in normal B cells [80]. In pre-clinical studies, CAR-T cells targeting ROR1 showed antitumor activity [81], although a phase 1 clinical trial of CAR-T cells targeting ROR1+ malignancies including CLL was recently terminated due to slow accrual (NCT02706392). Fc receptor for immunoglobulin M (FcuR) also has high expression in CLL and only minimal expression in normal B cells or other hematopoietic cells [82]. In pre-clinical studies, CAR-T cells targeting FcuR killed CLL cells while sparing normal B cells, suggesting another potential target for CAR-T cells in CLL [82].

In addition to evaluating new targets in CLL, there has also been interest in evaluating novel constructs to help overcome some of the challenges around CAR-T cell efficacy in CLL related to T cell quality as well as the inhibitory tumor microenvironment. One approach has been with armored CAR-T cells which are engineered to express additional proteins to augment their function. CD19 CAR-T cells with the CD28 costimulatory domain engineered to also express the 4-1BB ligand showed improved proliferation and persistence in pre-clinical models [83]. This design was further investigated in a phase 1 clinical trial in patients with relapsed/refractory NHL and CLL [84]. Twenty-five patients were treated, including nine patients with CLL and three patients with RT. The CR rate was 57%, including CRs in two out of nine CLL patients and one of two RT patients. Of note, most of the CLL patients were treated at lower dose levels. CAR-T cells were detected past 160 days. CRS was seen in 67% of patients (all grade 1 or 2) and neurotoxicity was reported in 33% of patients, with two patients presenting grade 3 neurotoxicity which was reversible.

### 4.3. Allogeneic CAR-T

Allogeneic CAR T-cell therapy (alloCAR-T), including donor-derived and off-the-shelf CAR-T cells, represent an emerging efficacious and safe treatment alternative to autologous CAR-T. AlloCAR-T offer a promising alternative as an off-the-shelf generalizable treatment product without the manufacturing difficulties that can be associated with autologous CAR-T. Donor derived CAR T-cells may be categorized as those obtained from donors in patients with history of prior alloHSCT or HLA-compatible donors, whereas off-the-shelf CAR T-cells are sourced from healthy unrelated donors [85]. Donor-derived alloCAR-T require HLA compatibility, whereas off-the-shelf CARs use techniques to minimize rejection of non-HLA matched products [86].

A number of studies have examined donor-derived alloCAR-T though primarily focused on B-ALL, with very few including CLL patients, given the limited role for alloHSCT in modern CLL management [87,88,89,90]. One phase 1 clinical trial included five CLL patients, with alloCAR-T were generated from the patient’s prior alloHSCT donor, and used a murine single-chain variable fragment antigen-recognition domain with a CD28 co-stimulatory domain and a CD3z T-cell activation domain [91]. Eight of the 20 patients who were treated with alloCAR-T attained CR and, furthermore, none of them developed new onset GVHD. Notably, a CLL patient achieved rapid elimination of disease after alloCAR-T, evidenced by a reduction of CLL cells from lymph nodes, blood, and bone marrow [91].

Off-the-shelf alloCAR-T are perhaps more relevant for modern CLL patients, though have primarily been studied in B-ALL and NHL. Two Phase 1 studies, including seven children and 14 adults with B-cell ALL, evaluated the safety profile and anti-leukemic activity of UCART19, an allogeneic genome edited anti-CD19 CAR T-cell product [92]. This product uses mRNA encoding transcription activator-like effector nucleases (TALEN) to disrupt the *CD52* gene to confer resistance to alemtuzumab that is used in lymphodepletion and to disrupt the *TRAC* gene to minimize GVHD risk. The most commonly reported adverse effects noted were CRS (91% patients, grade 3/4 in 14%), neurotoxicity (38%), grade 1 acute cutaneous GVHD (10%), and prolonged grade 4 cytopenias (32%). There were two treatment-related deaths, one from neutropenic sepsis with concurrent CRS and one from pulmonary hemorrhage in a patient with prolonged cytopenias. The CR/CRi rate was 67% but notably no UCART19 expansion or anti-leukemic efficacy was seen in the four patients who did not receive alemtuzumab [92].

Additional data for the adult relapsed or refractory B-cell ALL patients treated with UCART19 have recently been reported. For 25 patients, the ORR was 48% after a median follow up of 12.8 months. Three dose-limiting toxicities were seen, one at each dose level; one grade 4 CRS and two prolonged grade 4 cytopenia. None of the patients developed clinically significant GVHD, primarily attributed to the TCRαβ genome editing technique utilized in the development of UCART19 [86]. Although follow up is short, alloCAR-T may be a promising approach to apply in CLL given the limitations with autologous T-cell quality.

There are emerging techniques for alloCAR-T under development. One example is a gamma-delta alloCAR-T targeting CD20 (ADI-001) to harness both adaptive and innate cytotoxic effector functions, with responses seen in patients with LBCL and (NCT04735471) [93]. There was one RT patient on this trial who developed progressive disease on day 28. Another novel alloCAR-T construct (CB-010) employs CRISPR-based knockout of both *TRAC* and *PD-1* (intended to minimize CAR-T exhaustion), with site-specific insertion of CD19-specific CAR into the *TRAC* locus (NCT04637763). Six patients with NHL (no CLL/RT enrolled) have been dosed with five efficacy evaluable patients achieving responses at 4.5 months median follow up [94]. FT819 is an induced pluripotent stem cell (iPSC)-derived, CD19-targeting CAR-T with bi-allelic disruption of the T-cell receptor (TCR) to prevent GVHD, in addition to *TRAC* locus disruption and a novel 1XX CAR-signaling domain which extends T-cell effector function. In the 12 relapsed/refractory B-cell malignancy patients treated, safety but not efficacy has been reported, with no dose-limiting toxicities (DLTs), immune effector-cell-associated neurotoxicity syndrome (ICANS), or GVHD seen [95].

ALPHA (NCT03939026) is a phase 1 dose-escalation study for ALLO-501 (a TALEN edited allogeneic anti-CD19 CAR-T product) in relapsed/refractory B-cell lymphomas (including LBCL and FL). In the 46 treated patients, no cases of GVHD or DLTs were reported, with cytopenia being the most common adverse event. The six-month CR rate of 36.4% was similar to rates reported with autologous CAR T regimens, highlighting that ALLO-501 has manageable safety with encouraging efficacy [1], with possible improved efficacy among the four patients receiving consolidation doses (ORR of 100% and CR of 75%) [96].

ALPHA2 (NCT04416984) is a phase 1/2 clinical trial of ALLO-501A (ALLO-501 with eliminated rituximab recognition domain) in relapsed/refractory LBCL. Fifteen patients have been treated (12 evaluable so far), of whom six received a single dose with 50% showing CR; six received a consolidation dose (a second dose of ALLO-501A on Day 28, in patients with stable disease after the initial dose). ORR and CRR were 50%. In the consolidation cohort, the ORR and CR were 66.7% (with three PR converting to CR after the consolidation dose). Though follow up is short, these results represent encouraging evidence for the role of alloCAR-T with consolidative dosing [97].

PBCAR0191 is an alloCAR-T that uses a next-generation meganuclease platform called ARCUS for gene editing and has demonstrated efficacy in NHL and ALL patients, especially when intensified lymphodepletion is utilized [98,99]. The ARCUS platform is also used for PBCAR20A, an anti-CD20 alloCAR-T, which has completed recruitment for its phase 1/2a trial enrolling B-NHL and CLL, though results have not yet been reported (NCT04030195).

The use of alloCAR-T is not without its limitations. Determining the appropriate dose for alloCAR T-cells has proven to be a challenge, given that the cellular growth kinetics vary between patients and donors. Relapse, allo-reactivity, and donor T-cell rejection are among the other main issues that hamper their wider use in clinical trials. The limited persistence of alloCAR-T (typically weeks to months) in the recipient has been attributed as the primary reason for relapse. Studies indicate that persistence is likely impaired due to graft rejection by the host immune system or by immune suppressive medications intended to prevent GVHD.

In conclusion, alloCAR-T offer an emerging tool for advanced hematologic malignancies where use of an autologous CAR-T may not be feasible due to difficulty with production or rapid disease progression. A significant advantage for CLL patients may be improved T-cell quality compared to autologous-sourced products, though data from ongoing clinical trials, especially those including CLL patients, will be important to determine longer term safety and efficacy.

### 4.4. Natural Killer CARs

While CAR-T treatments have shown significant promise as a novel therapy in relapsed and refractory hematological malignancies, treatment can be limited by the logistical issues of autologous CAR-T production and the inherent risk of GVHD with alloCAR-T. Consequently, natural killer CARs (CAR NK) offer a potential alternative therapeutic strategy [100,101].

NK cells are part of innate immunity, with the ability to kill tumor cells without any prior activation or expression of MHC molecules, through cytotoxic secretion of perforin/granzyme granules that induce apoptosis [101]. In ALL, the presence of NK cells in the bone marrow was associated with better prognosis and response to chemotherapeutic agents [102]. NK cells rarely cause GvHD or CRS, making them suitable as the vehicle for an off-the-shelf generalized therapeutic product. NK cells can be sourced from donor blood cells, umbilical cord blood, or from induced pluripotent stem cells. One study reported that cord blood derived CAR NK engineered to express anti-CD19 CAR, IL-15, and inducible caspase 9 showed higher anti-tumor activity and higher rates of expansion in-vivo than non-transduced NK cells [103].

Some clinical trials are evaluating CAR NK in both solid tumors and lymphomas, whereas others focus on a single cancer type such as melanoma [104]. FT500 is an iPSC-derived NK cell therapy. A clinical trial showed that the majority of patients achieved stable disease though disease reduction was seen when combined with immune-checkpoint inhibition in one classical Hodgkin lymphoma patient [105,106].

Liu et al. from MD Andersons Cancer center conducted a Phase 1/2 trial at using HLA mismatched anti-CD19 CAR-NK, derived from cord blood and transduced with a retroviral vector expressing anti-CD19 CAR, IL-15, and inducible caspase 9, which were administered to 11 heavily pretreated patients with NHL (*n* = 6) and CLL/RT (*n* = 5) (median of four prior lines of therapy). There were no reported cases of CRS, neurotoxicity, or GVHD following the infusion. Eight patients went into complete remission, three with CLL, and one had remission of the RT component (but persistent CLL). At a median follow up of 13.8 months, two of three responding CLL patients required additional CLL therapy as did the RT patient [107].

Trogocytosis and subsequent fratricide may limit the efficacy of CAR-NK cells, though preclinical work using a dual-CAR system that adds an NK self-recognizing inhibitory signal may be one way to prevent this barrier [108]. CAR-NK are also hampered by limited persistence though the addition of IL-15 in CAR construction may help improve the lifespan [101,107].

Overall, early clinical trials suggest that CAR-NK could be a promising new tool in the arsenal of cellular therapies for CLL patients. However, further research will need to be conducted into engineering, transfection protocols, off-the-shelf production, and dose optimization.

### 4.5. CAR-T in Richter’s Transformation

Liso-cel has been utilized in RT, including in five patients from the phase 1/2 open-label trial where two attained CR, one PR, and two had progressive disease (PD). Moreover, five of the patients enrolled on the TRANSCEND NHL 001 trial had RT [41]. On efficacy analysis, RT patients were included with non-FL transformed indolent NHL patients, which also included those with marginal zone lymphoma (*n* = 10) and “other subtypes” (*n* = 3). The results showed a 61% ORR (11 of 18) and 39% CR. Kittai et al. also reported on the efficacy and safety of axi-cel in RT, based on institutional experience of off-label use in nine RT patients, of whom eight were evaluable [55]. The majority of patients continued with a BTK inhibitor during treatment. An ORR of 100% was seen with five CR and three PR as best response, with one progression event at six months median follow up. An early study using allogenic virus-specific T-cells that were engineered to express CD19 CAR included two RT patients, with one patient achieving PR followed by PD in two months [109].

## 5. Future Directions

Additional emerging techniques for the double-refractory CLL patient include use of a non-covalent BTKi [110,111], combined covalent BTKi and venetoclax [112,113], or other clinical trials including bispecifics, BTK degraders, etc. Therefore, cellular therapies, including CAR-T and CAR-NK, are among many potential therapies for this difficult-to-treat patient subset, for which ongoing efforts to improve safety and efficacy are of paramount importance (Figure 2). Given the improving safety of these newer agents, one would expect the role for alloHSCT to continue to diminish over time. However, RT remains a major area of unmet need with an ongoing need for improved therapies and expanded access. Moreover, further study of cellular therapies in this realm would likely benefit patients.

## 6. Conclusions

Cellular therapies have shown significant promise within the field of CLL and RT. Due to success of other novel agents, the role for alloHSCT is likely to continue to decline in CLL, though will still be important in RT given the paucity of therapies with longer term efficacy. CAR-T remain investigational for both CLL and RT though emerging technologies such as alloCAR-T and CAR NK may add to the existing cellular therapies toolset for these malignancies. 

## Figures and Tables

**Figure 1 cancers-15-01838-f001:**
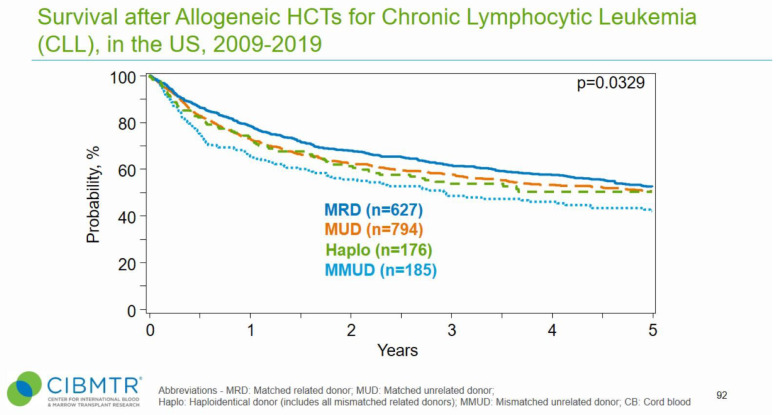
Survival following allogeneic hematopoietic stem cell transplants in the US. Source: CIBMTR.

**Figure 2 cancers-15-01838-f002:**
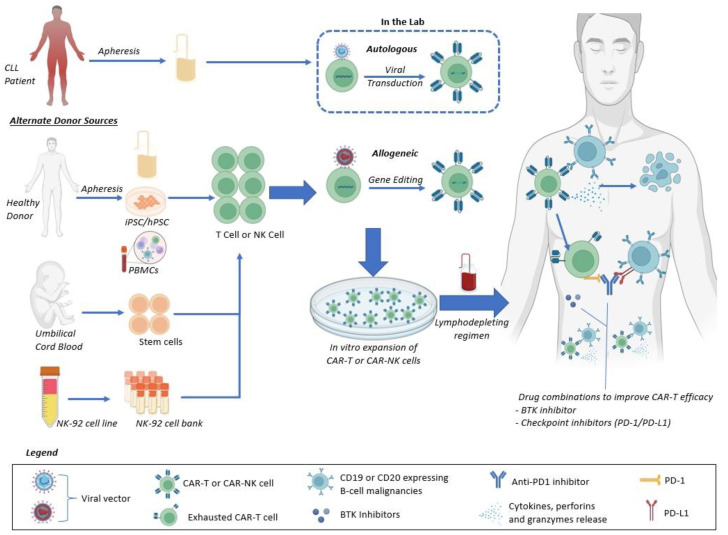
Schematic for CAR products with emphasis on emerging techniques to improve safety and efficacy in CLL patients. Allogeneic NK sources can include peripheral blood mononuclear cells (PBMCs), induced pluripotent stem cells (iPSC), human pluripotent stem cells (hPSC), NK cell lines, and umbilical cord blood. Allogeneic T-cell sources can include peripheral blood apheresis samples, iPSC/hPSC, and umblical cord blood.

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
