# Peer review of "Cellular Therapies in Chronic Lymphocytic Leukemia and Richter’s Transformation: Recent Developments in Chimeric Antigen Receptor T-Cells, Natural Killer Cells, and Allogeneic Stem Cell Transplant"

_cancers, 2023, doi:10.3390/cancers15061838_

Round 1

Reviewer 1 Report

Dear authors

The manuscript has been written well. The topic is of high importance and interesting. However, I propose to add the following references to the manuscript. 

Bahmanyar M, Vakil MK, Al-Awsi GR, Kouhpayeh SA, Mansoori Y, Mansoori B, Moravej A, Mazarzaei A, Ghasemian A. Anticancer traits of chimeric antigen receptors (CARs)-Natural Killer (NK) cells as novel approaches for melanoma treatment. BMC cancer. 2022 Dec;22(1):1-9.

Bahmanyar M, Vakil MK, Al-Awsi GR, Kouhpayeh SA, Mansoori H, Mansoori Y, Salahi A, Nikfar G, Tavassoli A, Behmard E, Moravej A. Opportunities and obstacles for the melanoma immunotherapy using T cell and chimeric antigen receptor T (CAR-T) applications: a literature review. Molecular Biology Reports. 2022 Nov;49(11):10627-33.

kind regards 

Author Response

The manuscript has been written well. The topic is of high importance and interesting. However, I propose to add the following references to the manuscript. 

Bahmanyar M, Vakil MK, Al-Awsi GR, Kouhpayeh SA, Mansoori Y, Mansoori B, Moravej A, Mazarzaei A, Ghasemian A. Anticancer traits of chimeric antigen receptors (CARs)-Natural Killer (NK) cells as novel approaches for melanoma treatment. BMC cancer. 2022 Dec;22(1):1-9.

Bahmanyar M, Vakil MK, Al-Awsi GR, Kouhpayeh SA, Mansoori H, Mansoori Y, Salahi A, Nikfar G, Tavassoli A, Behmard E, Moravej A. Opportunities and obstacles for the melanoma immunotherapy using T cell and chimeric antigen receptor T (CAR-T) applications: a literature review. Molecular Biology Reports. 2022 Nov;49(11):10627-33.

We thank the reviewer for their time and comments. We added the suggested references to our review along with a broad review on CAR-T in solid tumors for readers' interest:

End of first paragraph in introduction, new sentence:

CAR-T remain investigational for many other malignancies(Bahmanyar et al. 2022; Guzman et al. 2023). 

In section on CAR NK:

Some clinical trials are evaluating CAR NK in both solid tumors and lymphomas, whereas others focus on a single cancer type such as melanoma(Bahmanyar et al. 2022). 

Reviewer 2 Report

In this review by Coombs et al. the authors reviewed recent studies on CLL and RT and the treatments used for each, specifically on CAR-T therapies. The review outlines all the FDA approved treatments, new strategies to improve current CAR-T therapies, as well as new upcoming CAR-T therapies. It is a very well written review with up-to-date information on the current and new treatment modalities in CLL and RT.

The review is very thorough in all the topics covered and should be accepted with minor reviews.  Of note, Figure 2 should be revised. There is a square where the characters are not legible and should be changed. In addition, in Figure 2, the figure could be changed to make it clearer. There should be an arrow from the CLL patient - apheresis directly to Autologous. Leaving the arrow from Alternate Donor Sources directly to Allogeneic.

Author Response

We thank the reviewer for these comments.

Regarding Figure 2, we appreciate the attention to detail - to improve clarity, we have removed the text in the prior version's lower left corner and this is now in the figure description. We slightly increased the size of remaining font and used an upgraded version of the software we used to make the figure to improve image quality. We also added the arrow as suggested.